# META-GRAPH: FEW SHOT LINK PREDICTION VIA META LEARNING

## ABSTRACT

We consider the task of *few shot link prediction*, where the goal is to predict missing edges across multiple graphs using only a small sample of known edges. We show that current link prediction methods are generally ill-equipped to handle this task—as they cannot effectively transfer knowledge between graphs in a multigraph setting and are unable to effectively learn from very sparse data. To address this challenge, we introduce a new gradient-based meta learning framework, *Meta-Graph*, that leverages higher-order gradients along with a learned graph signature function that conditionally generates a graph neural network initialization. Using a novel set of few shot link prediction benchmarks, we show that *Meta-Graph* enables not only fast adaptation but also better final convergence and can effectively learn using only a small sample of true edges.

## 1  INTRODUCTION

Given a graph representing known relationships between a set of nodes, the goal of link prediction is to learn from the graph and infer novel or previously unknown relationships (Liben-Nowell & Kleinberg, 2003). For instance, in a social network we may use link prediction to power a friendship recommendation system (Aiello et al., 2012), or in the case of biological network data we might use link prediction to infer possible relationships between drugs, proteins, and diseases (Zitnik & Leskovec, 2017). However, despite its popularity, previous work on link prediction generally focuses only on one particular problem setting: it generally assumes that link prediction is to be performed on a single large graph and that this graph is relatively complete, i.e., that at least 50% of the true edges are observed during training (e.g., see Grover & Leskovec, 2016; Kipf & Welling, 2016b; Liben-Nowell & Kleinberg, 2003; Lü & Zhou, 2011).

In this work, we consider the more challenging setting of *few shot link prediction*, where the goal is to perform link prediction on multiple graphs that contain only a small fraction of their true, underlying edges. This task is inspired by applications where we have access to multiple graphs from a single domain but where each of these individual graphs contains only a small fraction of the true, underlying edges. For example, in the biological setting, high-throughput interactomics offers the possibility to estimate thousands of biological interaction networks from different tissues, cell types, and organisms (Barrios-Rodiles et al., 2005); however, these estimated relationships can be noisy and sparse, and we need learning algorithms that can leverage information across these multiple graphs in order to overcome this sparsity. Similarly, in the e-commerce and social network settings, link prediction can often have a large impact in cases where we must quickly make predictions on sparsely-estimated graphs, such as when a service has been recently deployed to a new locale. That is to say to link prediction for a new sparse graph can benefit from transferring knowledge from other, possibly more dense, graphs assuming there is exploitable shared structure.

We term this problem of link prediction from sparsely-estimated multi-graph data as few shot link prediction analogous to the popular few shot classification setting (Miller et al., 2000; Lake et al., 2011; Koch et al., 2015). The goal of few shot link prediction is to observe many examples of graphs from a particular domain and leverage this experience to enable fast adaptation and higher accuracy when predicting edges on a new, sparsely-estimated graph from the same domain—a task that can can also be viewed as a form of meta learning, or learning to learn (Bengio et al., 1990; 1992; Thrun & Pratt, 2012; Schmidhuber, 1987) in the context of link prediction. This few shot link prediction setting is particularly challenging as current link prediction methods are generally ill-equipped to

transfer knowledge between graphs in a multi-graph setting and are also unable to effectively learn from very sparse data.

**Present work**. We introduce a new framework called *Meta-Graph* for few shot link prediction and also introduce a series of benchmarks for this task. We adapt the classical gradient-based meta-learning formulation for few shot classification (Miller et al., 2000; Lake et al., 2011; Koch et al., 2015) to the graph domain. Specifically, we consider a distribution over graphs as the distribution over tasks from which a global set of parameters are learnt, and we deploy this strategy to train graph neural networks (GNNs) that are capable of few-shot link prediction. To further bootstrap fast adaptation to new graphs we also introduce a graph signature function, which learns how to map the structure of an input graph to an effective initialization point for a GNN link prediction model. We experimentally validate our approach on three link prediction benchmarks. We find that our Meta-Graph approach not only achieves fast adaptation but also converges to a better overall solution in many experimental settings, with an average improvement of $5.3\%$ in AUC at convergence over non-meta learning baselines.

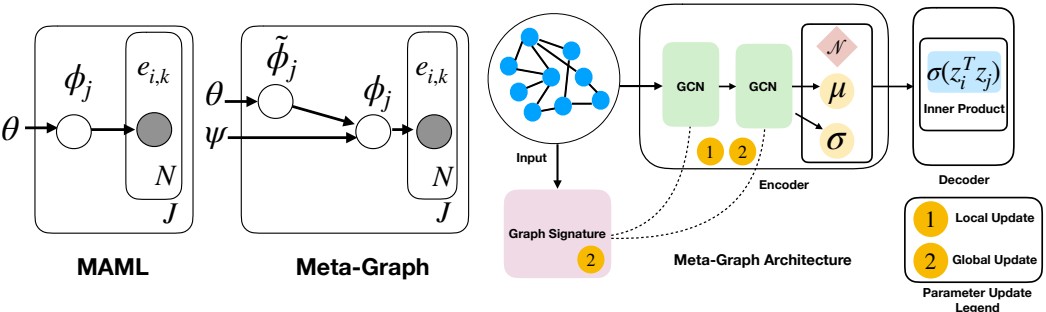

Figure 1: **Left:** Graphical model for Meta-Graph vs. MAML. **Right:** Meta-Graph architecture.

## 2 PRELIMINARIES AND PROBLEM DEFINITION

The basic set-up for few shot link prediction is as follows: We assume that we have a distribution $p(\mathcal{G})$ over graphs, from which we can sample training graphs $\mathcal{G}_i \sim p(\mathcal{G})$, where each $\mathcal{G}_i = (\mathcal{V}_i, \mathcal{E}_i, X_i)$ is defined by a set of nodes $\mathcal{V}_i$, edges $\mathcal{E}_i$, and matrix of real-valued node attributes $X \in \mathbb{R}^{|\mathcal{V}_i| \times d}$. When convenient, we will also equivalently represent a graph as $\mathcal{G}_i = (\mathcal{V}_i, A_i, X_i)$, where $A_i \in \mathbb{Z}^{|\mathcal{V}_i| \times |\mathcal{V}_i|}$ is an adjacency matrix representation of the edges in $\mathcal{E}_i$. We assume that each of these sampled graphs, $\mathcal{G}_i$, is a simple graph (i.e., contain a single type of relation and no self loops) and that every node $v \in \mathcal{V}_i$ in the graph is associated with a real valued attribute vector $\mathbf{x}_v \in \mathbb{R}^d$ from a common vector space. We further assume that for each graph $\mathcal{G}_i$ we have access to only a sparse subset of the true edges $\mathcal{E}_i^{\text{train}} \subset \mathcal{E}_i$ (with $|\mathcal{E}_i^{\text{train}}| << |\mathcal{E}_i|$) during training. In terms of distributional assumptions we assume that this $p(\mathcal{G})$ is defined over a set of related graphs (e.g., graphs drawn from a common domain or application setting).

Our goal is to learn a *global* or *meta* link prediction model from a set of sampled training graphs $\mathcal{G}_i \sim p(\mathcal{G}), i = 1...n$, such that we can use this meta model to quickly learn an effective link prediction model on a newly sampled graph $\mathcal{G}_* \sim p(\mathcal{G})$. More specifically, we wish to optimize a global set of parameters $\theta$, as well as a graph signature function $\psi(\mathcal{G}_i)$, which can be used together to generate an effective parameter initialization, $\phi_i$, for a *local* link prediction model on graph $\mathcal{G}_i$.

**Relationship to standard link prediction**. Few shot link prediction differs from standard link prediction in three important ways:

1. Rather than learning from a single graph $\mathcal{G}$, we are learning from multiple graphs $\{\mathcal{G}_1, ..., \mathcal{G}_n\}$ sampled from a common distribution or domain.
2. We presume access to only a very sparse sample of true edges. Concretely, we focus on settings where at most 30% of the edges in $\mathcal{E}_i$ are observed during training, i.e., where $\frac{|\mathcal{E}^{\text{train}}|}{|\mathcal{E}|} \leq 0.3$.[1]

---

[1]By "true edges" we mean the full set of ground truth edges available in a particular dataset.

3. We distinguish between the *global* parameters, which are used to encode knowledge about the underlying distribution of graphs, and the *local* parameters $\phi_i$, which are optimized to perform link prediction on a specific graph $\mathcal{G}_i$. This distinction allows us to consider leveraging information from multiple graphs, while still allowing for individually-tuned link prediction models on each specific graph.

**Relationship to traditional meta learning**. Traditional meta learning for few-shot classification, generally assumes a distribution $p(\mathcal{T})$ over classification tasks, with the goal of learning global parameters that can facilitate fast adaptation to a newly sampled task $\mathcal{T}_i \sim p(\mathcal{T})$ with few examples. We instead consider a distribution $p(\mathcal{G})$ over graphs with the goal of performing link prediction on a newly sampled graph. An important complication of this graph setting is that the individual predictions for each graph (i.e., the training edges) are not i.i.d.. Furthermore, for few shot link prediction we require training samples as a sparse subset of true edges that represents a small percentage of all edges in a graph. Note that for very small percentages we effectively break all graph structure and recover the supervised setting for few shot classification and thus simplifying the problem.

## 3 Proposed Approach

We now outline our proposed approach, Meta-Graph, to the few shot link prediction problem. We first describe how we define the local link prediction models, which are used to perform link prediction on each specific graph $\mathcal{G}_i$. Next, we discuss our novel gradient-based meta learning approach to define a global model that can learn from multiple graphs to generate effective parameter initializations for the local models. The key idea behind Meta-Graph is that we use gradient-based meta learning to optimize a shared parameter initialization $\theta$ for the local models, while also learning a parametric encoding of each graph $\mathcal{G}_i$ that can be used to modulate this parameter initialization in a graph-specific way (Figure 1).

### 3.1 Local Link Prediction Model

In principle, our framework can be combined with a wide variety of GNN-based link prediction approaches, but here we focus on variational graph autoencoders (VGAEs) (Kipf & Welling, 2016b) as our base link prediction framework. Formally, given a graph $\mathcal{G} = (\mathcal{V}, A, X)$, the VGAE learns an inference model, $q_\phi$, that defines a distribution over node embeddings $q_\phi(Z|A, X)$, where each row $z_v \in \mathbb{R}^d$ of $Z \in \mathbb{R}^{|\mathcal{V}| \times d}$ is a node embedding that can be used to score the likelihood of an edge existing between pairs of nodes. The parameters of the inference model are shared across all the nodes in $\mathcal{G}$, to define the approximate posterior $q_\phi(z_v|A, X) = \mathcal{N}(z_v|\mu_v, \text{diag}(\sigma_v^2))$, where the parameters of the normal distribution are learned via GNNs:

$$\mu = \text{GNN}_\mu(A, X), \qquad \text{and} \qquad \log(\sigma) = \text{GNN}_\sigma(A, X). \tag{1}$$

The generative component of the VGAE is then defined as

$$p(A|Z) = \prod_{i=1}^{N} \prod_{j=1}^{N} p(A_{u,v}|z_u, z_v), \qquad \text{with} \qquad p(A_{u,v}|z_u, z_v) = \sigma(z_u^\top z_v), \tag{2}$$

i.e., the likelihood of an edge existing between two nodes, $u$ and $v$, is proportional to the dot product of their node embeddings. Given the above components, the inference GNNs can be trained to minimize the variational lower bound on the training data:

$$\mathcal{L}_G = \mathbb{E}_{q_\phi}[\log p(A^{\text{train}}|Z)] - KL[q_\phi(Z|X, A^{\text{train}})||p(z)], \tag{3}$$

where a Gaussian prior is used for $p(z)$.

We build upon VGAEs due to their strong performance on standard link prediction benchmarks (Kipf & Welling, 2016b), as well as the fact that they have a well-defined probabilistic interpretation that generalizes many embedding-based approaches to link prediction (e.g., node2vec (Grover & Leskovec, 2016)). We describe the specific GNN implementations we deploy for the inference model in Section 3.3.

## 3.2 Overview of Meta-Graph

The key idea behind Meta-Graph is that we use gradient-based meta learning to optimize a shared parameter initialization $\theta$ for the inference models of a VGAE, while also learning a parametric encoding $\psi(\mathcal{G}_i)$ that modulates this parameter initialization in a graph-specific way. Specifically, given a sampled training graph $\mathcal{G}_i$, we initialize the inference model $q_{\phi_i}$ for a VGAE link prediction model using a combination of two learned components:

- A global initialization, $\theta$, that is used to initialize all the parameters of the GNNs in the inference model. The global parameters $\theta$ are optimized via second-order gradient descent to provide an effective initialization point for any graph sampled from the distribution $p(\mathcal{G})$.
- A graph signature $s_{\mathcal{G}_i} = \psi(\mathcal{G}_i)$ that is used to modulate the parameters of inference model $\phi_i$ based on the history of observed training graphs. In particular, we assume that the inference model $q_{\phi_i}$ for each graph $\mathcal{G}_i$ can be conditioned on the graph signature. That is, we augment the inference model to $q_{\phi_i}(Z|A, X, s_{\mathcal{G}_i})$, where we also include the graph signature $s_{\mathcal{G}_i}$ as a conditioning input. We use a k-layer graph convolutional network (GCN) (Kipf & Welling, 2016a), with sum pooling to compute the signature:

$$s_{\mathcal{G}} = \psi(\mathcal{G}) = \text{MLP}(\sum_{v \in \mathcal{V}} z_v) \qquad \text{with} \qquad Z = \text{GCN}(A, X), \qquad (4)$$

where GCN denotes a k-layer GCN (as defined in (Kipf & Welling, 2016a)), MLP denotes a densely-connected neural network, and we are summing over the node embeddings $z_v$ output from the GCN. As with the global parameters $\theta$, the graph signature model $\psi$ is optimized via second-order gradient descent.

The overall Meta-Graph architecture is detailed in Figure 1 and the core learning algorithm is summarized in the algorithm block below.

---

**Algorithm 1:** Meta-Graph for Few Shot Link Prediction

---

**Result:** Global parameters $\theta$, Graph signature function $\psi$
Initialize learning rates: $\alpha, \epsilon$
Sample a mini-batch of graphs, $\mathcal{G}_{batch}$ from $p(\mathcal{G})$;
**for** *each* $\mathcal{G} \in \mathcal{G}_{batch}$ **do**
    $\mathcal{E} = \mathcal{E}^{\text{train}} \cup \mathcal{E}^{\text{val}} \cup \mathcal{E}^{\text{test}}$ // Split edges into train, val, and test
    $s_{\mathcal{G}} = \psi(\mathcal{G}, \mathcal{E}^{\text{train}})$ // Compute graph signature
    Initialize: $\phi^{(0)} \leftarrow \theta$ // Initialize local parameters via global parameters
    **for** $k$ *in* $[1 : K]$ **do**
        $s_{\mathcal{G}} = \text{stopgrad}(s_{\mathcal{G}})$ // Stop Gradients to Graph Signature
        $\mathcal{L}_{train} = \mathbb{E}_q[\log p(A^{\text{train}}|Z)] - KL[q_\phi(Z|\mathcal{E}^{\text{train}}, s_{\mathcal{G}})||p(z)]$
        Update $\phi^{(k)} \leftarrow \phi^{(k-1)} - \alpha \nabla_\phi \mathcal{L}_{train}$
    **end**
    Initialize: $\theta \leftarrow \phi_K$
    $s_{\mathcal{G}} = \psi(\mathcal{G}, \mathcal{E}^{\text{val}} \cup \mathcal{E}^{\text{train}})$ // Compute graph signature with validation edges
    $\mathcal{L}_{val} = \mathbb{E}_q[\log p(A^{\text{val}}|Z)] - KL[q(Z|\mathcal{E}^{\text{val}} \cup \mathcal{E}^{\text{train}}, s_{\mathcal{G}})||p(z)]$
    Update $\theta \leftarrow \theta - \epsilon \nabla_\theta \mathcal{L}_{val}$
    Update $\psi \leftarrow \psi - \epsilon \nabla_\psi \mathcal{L}_{val}$
**end**

---

The basic idea behind the algorithm is that we (i) sample a batch of training graphs, (ii) initialize VGAE link prediction models for these training graphs using our global parameters and signature function, (iii) run $K$ steps of gradient descent to optimize each of these VGAE models, and (iv) use second order gradient descent to update the global parameters and signature function based on a held-out validation set of edges. As depicted in Fig 1, this corresponds to updating the GCN based encoder for the local link prediction parameters $\phi_j$ and global parameters $\theta$ along with the graph signature function $\psi$ using second order gradients. Note that since we are running $K$ steps of gradient descent within the inner loop of Algorithm 1, we are also "meta" optimizing for fast adaptation, as $\theta$ and $\psi$ are being trained via second-order gradient descent to optimize the local model performance after $K$ gradient updates, where generally $K \in \{0, 1, \ldots, 5\}$.

### 3.3 Variants of Meta-Graph

We consider several concrete instantiations of the Meta-Graph framework, which differ in terms of how the output of the graph signature function is used to modulate the parameters of the VGAE inference models. For all the Meta-Graph variants, we build upon the standard GCN propagation rule (Kipf & Welling, 2016a) to construct the VGAE inference models. In particular, we assume that all the inference GNNs (Equation 1) are defined by stacking $K$ neural message passing layers of the form:

$$h_v^{(k)} = \text{ReLU} \left( \sum_{u \in \mathcal{N}(v) \cup \{v\}} \frac{m_{s_\mathcal{G}} \left( W^{(k)} h_u^{(k-1)} \right)}{\sqrt{|\mathcal{N}(v)||\mathcal{N}(u)|}} \right), \tag{5}$$

where $h_v \in \mathbb{R}^d$ denotes the embedding of node $v$ at layer $k$ of the model, $\mathcal{N}(v) = \{u \in \mathcal{V} : e_{u,v} \in \mathcal{E}\}$ denotes the nodes in the graph neighborhood of $v$, and $W^{(k)} \in \mathbb{R}^{d \times d}$ is a trainable weight matrix for layer $k$. The key difference between Equation 5 and the standard GCN propagation rule is that we add the modulation function $m_{s_\mathcal{G}}$, which is used to modulate the message passing based on the graph signature $s_\mathcal{G} = \psi(\mathcal{G})$.

We describe different variations of this modulation below. In all cases, the intuition behind this modulation is that we want to compute a structural signature from the input graphs that can be used to condition the initialization of the local link prediction models. Intuitively, we expect this graph signature to encode structural properties of sampled graphs $\mathcal{G}_i \sim p(\mathcal{G})$ in order to modulate the parameters of the local VGAE link prediction models and adapt it to the current graph.

**GS-Modulation**. Inspired by Brockschmidt (2019), we experiment with basic feature-wise linear modulation (Strub et al., 2018) to define the modulation function $m_{s_\mathcal{G}}$:

$$\beta_k, \gamma_k, = \psi(\mathcal{G})$$
$$m_{\beta_k, \gamma_k} \left( W^{(k)} h_u^{(k-1)} \right) = \gamma_k \odot W h^{(k-1)} + \beta_k. \tag{6}$$

Here, we restrict the modulation terms $\beta_k$ and $\gamma_k$ output by the signature function to be in $[-1, 1]$ by applying a $\tanh$ non-linearity after Equation 4.

**GS-Gating**. Feature-wise linear modulation of the GCN parameters (Equation 6) is an intuitive and simple choice that provides flexible modulation while still being relatively constrained. However, one drawback of the basic linear modulation is that it is "always on", and there may be instances where the modulation could actually be counter-productive to learning. To allow the model to adaptively learn when to apply modulation, we extend the feature-wise linear modulation using a sigmoid gating term, $\rho_k$ (with $[0, 1]$ entries), that gates in the influence of $\gamma$ and $\beta$:

$$\beta_k, \gamma_k, \rho_k = \psi(\mathcal{G})$$
$$\beta_k = \rho_k \odot \beta_k + (\mathbb{1} - \rho_k) \odot \mathbb{1}$$
$$\gamma_k = \rho_k \odot \gamma_k + (\mathbb{1} - \rho_k) \odot \mathbb{1}$$
$$m_{\beta_k, \gamma_k} \left( W^{(k)} h_u^{(k-1)} \right) = \gamma_k \odot W h^{(k-1)} + \beta_k.$$

**GS-Weights**. In the final variant of Meta-Graph, we extend the gating and modulation idea by separately aggregating graph neighborhood information with and without modulation and then merging these two signals via a convex combination:

$$\beta_k, \gamma_k, \rho_k = \psi(\mathcal{G})$$
$$h_v^{(k),1} = \text{ReLU} \left( \sum_{u \in \mathcal{N}(v) \cup \{v\}} \frac{W^{(k)} h_u^{(k-1)}}{\sqrt{|\mathcal{N}(v)||\mathcal{N}(u)|}} \right)$$
$$h_v^{(k),2} = \text{ReLU} \left( \sum_{u \in \mathcal{N}(v) \cup \{v\}} \frac{m_{s_{\beta_k, \gamma_k}} \left( W^{(k)} h_u^{(k-1)} \right)}{\sqrt{|\mathcal{N}(v)||\mathcal{N}(u)|}} \right)$$
$$h_v^{(k)} = \rho_k \odot h_v^{(k),1} + (\mathbb{1} - \rho_k) \odot h_v^{(k),2},$$

where we use the basic linear modulation (Equation 6) to define $m_{s_{\beta_k, \gamma_k}}$.

### 3.4 MAML FOR LINK PREDICTION AS A SPECIAL CASE

Note that a simplification of Meta-Graph, where the graph signature function is removed, can be viewed as an adaptation of model agnostic meta learning (MAML) (Finn et al., 2017) to the few shot link prediction setting. As discussed in Section 2, there are important differences in the set-up for few shot link prediction, compared to traditional few shot classification. Nonetheless, the core idea of leveraging an inner and outer loop of training in Algorithm 1—as well as using second order gradients to optimize the global parameters—can be viewed as an adaptation of MAML to the graph setting, and we provide comparisons to this simplified MAML approach in the experiments below. We formalize the key differences by depicting the graphical model of MAML as first depicted in (Grant et al., 2018) and contrasting it with the graphical model for Meta-Graph, in Figure 1. MAML when reinterpreted for a distribution over graphs, maximizes the likelihood over all edges in the distribution. On the other hand, Meta-Graph when recast in a hierarchical Bayesian framework adds a graph signature function that influences $\tilde{\phi}_j$ to produce the modulated parameters $\phi_j$ from $N$ sampled edges. This explicit influence of $\psi$ is captured by the term $p(\tilde{\phi}_j|\psi, \phi_j)$ in Equation 7 below:

$$p(\mathcal{E}|\theta) = \prod_j^J \left( \int \int p(\mathcal{E}_j|\phi_j)p(\phi_j|\psi, \tilde{\phi}_j)p(\tilde{\phi}_j|\theta)d\phi_j d\tilde{\phi}_j \right) \tag{7}$$

For computational tractability we take the likelihood of the modulated parameters as a point estimate —i.e., $p(\phi_j|\psi, \tilde{\phi}_j) = \delta(\psi \cdot \tilde{\phi}_j)$.

## 4 EXPERIMENTS

We design three novel benchmarks for the few-shot link prediction task. All of these benchmarks contain a set of graphs drawn from a common domain. In all settings, we use 80% of these graphs for training and 10% as validation graphs, where these training and validation graphs are used to optimize the global model parameters (for Meta-Graph) or pre-train weights (for various baseline approaches). We then provide the remaining 10% of the graphs as test graphs, and our goal is to fine-tune or train a model on these test graphs to achieve high link prediction accuracy. Note that in this few shot link prediction setting, *there are train/val/test splits at both the level of graphs and edges*: for every individual graph, we are optimizing a model using the training edges to predict the likelihood of the test edges, but we are also training on multiple graphs with the goal of facilitating fast adaptation to new graphs via the global model parameters.

Our goal is to use our benchmarks to investigate four key empirical questions:

**Q1** How does the overall performance of Meta-Graph compare to various baselines, including (i) a simple adaptation of MAML (Finn et al., 2017) (i.e., an ablation of Meta-Graph where the graph signature function is removed), (ii), standard pre-training approaches where we pre-train the VGAE model on the training graphs before fine-tuning on the test graphs, and (iii) naive baselines that do not leverage multi-graph information (i.e., a basic VGAE without pre-training, the Adamic-Adar heuristic (Adamic & Adar, 2003), and DeepWalk (Perozzi et al., 2014))?

**Q2** How well does Meta-Graph perform in terms of fast adaption? Is Meta-Graph able to achieve strong performance after only a small number of gradient steps on the test graphs?

**Q3** How necessary is the graph signature function for strong performance, and how do the different variants of the Meta-Graph signature function compare across the various benchmark settings?

**Q4** What is learned by the graph signature function? For example, do the learned graph signatures correlate with the structural properties of the input graphs, or are they more sensitive to node feature information?

**Datasets**. Two of our benchmarks are derived from standard multi-graph datasets from protein-protein interaction (PPI) networks (Zitnik & Leskovec, 2017) and 3D point cloud data (FirstMM-DB) (Neumann et al., 2013). These benchmarks are traditionally used for node and graph classification, respectively, but we adapt them for link prediction. We also create a novel multi-graph dataset based upon the AMINER citation data (Tang et al., 2008), where each node corresponds to a paper and links represent citations. We construct individual graphs from AMINER data by sampling ego networks around nodes and create node features using embeddings of the paper abstracts (see Appendix for details). We preprocess all graphs in each domain such that each graph contains a

Table 1: Statistics for the three datasets used to test Meta-Graph.

| DATASET | #GRAPHS | AVG. NODES | AVG. EDGES | #NODE FEATS |
|---|---|---|---|---|
| PPI | 24 | 2,331 | 64,596 | 50 |
| FIRSTMM DB | 41 | 1,377 | 6,147 | 5 |
| EGO-AMINER | 72 | 462 | 2245 | 300 |

| | PPI | | | FirstMM DB | | | Ego-AMINER | | |
|---|---|---|---|---|---|---|---|---|---|
| Edges | 10% | 20% | 30% | 10% | 20% | 30% | 10% | 20% | 30% |
| Meta-Graph | **0.795** | **0.833** | **0.845** | **0.782** | **0.786** | 0.783 | **0.626** | **0.738** | **0.786** |
| MAML | 0.770 | 0.815 | 0.828 | 0.776 | 0.782 | **0.793** | 0.561 | 0.662 | 0.667 |
| Random | 0.578 | 0.651 | 0.697 | 0.742 | 0.732 | 0.720 | 0.500 | 0.500 | 0.500 |
| No Fintune | 0.738 | 0.786 | 0.801 | 0.740 | 0.710 | 0.734 | 0.548 | 0.621 | 0.673 |
| Finetune | 0.752 | 0.801 | 0.821 | 0.752 | 0.735 | 0.723 | 0.623 | 0.691 | 0.723 |
| Adamic | 0.540 | 0.623 | 0.697 | 0.504 | 0.519 | 0.544 | 0.515 | 0.549 | 0.597 |
| Deepwalk | 0.664 | 0.673 | 0.694 | 0.487 | 0.473 | 0.510 | 0.602 | 0.638 | 0.672 |

Table 2: Convergence AUC results for different training edge splits.

minimum of 100 nodes and up to a maximum of 20000 nodes. For all datasets, we perform link prediction by training on a small subset (i.e., a percentage) of the edges and then attempting to predict the unseen edges (with 20% of the held-out edges used for validation). Key dataset statistics are summarized in Table 1.

**Baseline details**. Several baselines correspond to modifications or ablations of Meta-Graph, including the straightforward adaptation of MAML (which we term *MAML* in the results), a finetune baseline where we pre-train a VGAE on the training graphs observed in a sequential order and finetune on the test graphs (termed *Finetune*). We also consider a VGAE trained individually on each test graph (termed *No Finetune*). For Meta-Graph and all of these baselines we employ Bayesian optimization with Thompson sampling (Kandasamy et al., 2018) to perform hyperparameter selection using the validation sets. We use the recommended default hyperparameters for DeepWalk and Adamic-Adar baseline is hyperparameter-free. [2]

## 4.1 RESULTS

**Q1: Overall Performance**. Table 2 shows the link prediction AUC for Meta-Graph and the baseline models when trained to convergence using 10%, 20% or 30% of the graph edges. In this setting, we adapt the link prediction models on the test graphs until learning converges, as determined by performance on the validation set of edges, and we report the average link prediction AUC over the test edges of the test graphs. Overall, we find that Meta-Graph achieves the highest average AUC in all but one setting, with an average relative improvement of 4.8% in AUC compared to the MAML approach and an improvement of 5.3% compared to the Finetune baseline. Notably, Meta-Graph is able to maintain especially strong performance when using only 10% of the graph edges for training, highlighting how our framework can learn from very sparse samples of edges. Interestingly, in the Ego-AMINER dataset, unlike PPI and FIRSTMM DB, we observe the relative difference in performance between Meta-Graph and MAML to increase with density of the training set. We hypothesize that this is due to fickle nature of optimization with higher order gradients in MAML (Antoniou et al., 2018) which is somewhat alleviated in GS-gating due to the gating mechanism. With respect to computational complexity we observe a slight overhead when comparing Meta-Graph to MAML which can be reconciled by realizing that the graph signature function is not updated in the inner loop update but only in outer loop. In the Appendix, we provide additional results when using larger sets of training edges, and, as expected, we find that the relative gains of Meta-Graph decrease as more and more training edges are available.

**Q2: Fast Adaptation**. Table 3 highlights the average AUCs achieved by Meta-Graph and the baselines after performing only 5 gradient updates on the batch of training edges. Note that in this

---

[2]Code is included with our submission and will be made public after the review process

| | PPI | | | FirstMM DB | | | Ego-AMINER | | |
|---|---|---|---|---|---|---|---|---|---|
| Edges | 10% | 20% | 30% | 10% | 20% | 30% | 10% | 20% | 30% |
| Meta-Graph | **0.795** | **0.824** | **0.847** | **0.773** | **0.767** | 0.737 | **0.620** | 0.585 | **0.732** |
| MAML | 0.728 | 0.809 | 0.804 | 0.763 | 0.750 | **0.750** | 0.500 | 0.504 | 0.500 |
| No Fintune | 0.600 | 0.697 | 0.717 | 0.708 | 0.680 | 0.709 | 0.500 | 0.500 | 0.500 |
| Finetune | 0.582 | 0.727 | 0.774 | 0.705 | 0.695 | 0.704 | 0.608 | **0.675** | 0.713 |

Table 3: 5-gradient update AUC results with various fractions of training edges.

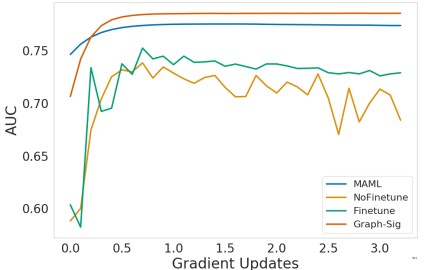 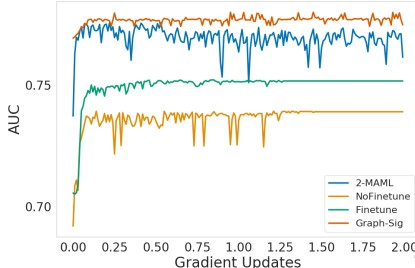

Figure 2: AUC scores on PPI (**Left**) and FirstMM DB (**Right**) graphs with $10\%$ of edges observed.

setting we only compare to the MAML, Finetune, and No Finetune baselines, as fast adaption in this setting is not well defined for the DeepWalk and Adamic-Adar baselines. In terms of fast adaptation, we again find that Meta-Graph is able to outperform all the baselines in all but one setting, with an average relative improvement of $9.4\%$ compared to MAML and $8.0\%$ compared to the Finetune baseline—highlighting that Meta-Graph can not only learn from sparse samples of edges but is also able to quickly learn on new data using only a small number of gradient steps. Also, we observe poor performance for MAML in the Ego-AMINER dataset dataset which we hypothesize is due to extremely low learning rates —i.e. $1e - 7$ needed for any learning, the addition of a graph signature alleviates this problem. Figure 2 shows the learning curves for the various models on the PPI and FirstMM DB datasets, where we can see that Meta-Graph learns very quickly but can also begin to overfit after only a small number of gradient updates, making early stopping essential.

**Q3: Choice of Meta-Graph Architecture**. We study the impact of the graph signature function and its variants GS-Gating and GS-Weights by performing an ablation study using the FirstMM DB dataset. Figure 3 shows the performance of the different model variants and baselines considered as the training progresses. In addition to models that utilize different signature functions we report a random baseline where parameters are initialized but never updated allowing us to assess the inherent power of the VGAE model for few-shot link prediction. To better understand the utility of using a GCN based inference network we also report a VGAE model that uses a simple MLP on the node features and is trained analogously to Meta-Graph as a baseline. As shown in Figure 3 many versions of the signature function start at a better initialization point or quickly achieve higher AUC scores in comparison to MAML and the other baselines, but simple modulation and GS-Gating are superior to GS-Weights after a few gradient steps.

**Q4: What is learned by the graph signature?** To gain further insight into what knowledge is transferable among graphs we use the FirstMM DB and Ego-AMINER datasets to probe and compare the output of the signature function with various graph heuristics. In particular, we treat the output of $s_{\mathcal{G}} = \psi(\mathcal{G})$ as a vector and compute the cosine similarity between all pairs of graph in the training set (i.e., we compute the pairwise cosine similarites between graph signatures, $s_{\mathcal{G}}$). We similarly compute three pairwise graph statistics—namely, the cosine similarity between average node features in the graphs, the difference in number of nodes, and the difference in number of edges—and we compute the Pearson correlation between the pairwise graph signature similarities and these other pairwise statistics. As shown in Table 4 we find strong positive correlation in terms of Pearson correlation coefficient between node features and the output of the signature function for both datasets, indicating that the graph signature function is highly sensitive to feature information. This observation is not entirely surprising given that we use such sparse samples of edges—meaning that many structural graph properties are likely lost and making the meta-learning heavily reliant on node feature information. We also observe moderate negative correlation with respect to the average

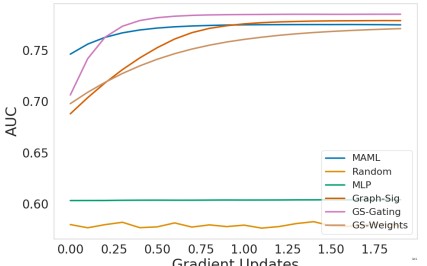 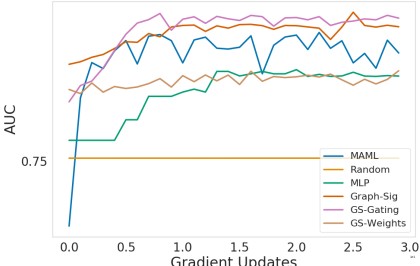

Figure 3: Ablation study on PPI **(Left)** and FirstMM DB **(Right)** graphs with $10\%$ of edges.

|  | FirstMM DB | | | Ego-AMINER | | |
|---|---|---|---|---|---|---|
| % Edges | 10% | 20% | 30% | 10% | 20% | 30% |
| Node Feats | 0.928 | 0.950 | 0.761 | 0.473 | 0.385 | 0.448 |
| Diff Num. Nodes | -0.093 | -0.196 | -0.286 | 0.095 | 0.086 | 0.085 |
| Diff Num. Edges | -0.093 | -0.195 | -0.281 | 0.093 | 0.072 | 0.075 |

Table 4: Pearson scores between graph signature output and other graph statistics.

difference in nodes and edges between pairs of graphs for FirstMM DB dataset. For Ego-AMINER we observe small positive correlation for difference in nodes and edges.

## 5 RELATED WORK

We now briefly highlight related work on link prediction, meta-learning, few-shot classification, and few-shot learning in knowledge graphs. Link prediction considers the problem of predicting missing edges between two nodes in a graph that are likely to have an edge. (Liben-Nowell & Kleinberg, 2003). Common successful applications of link prediction include friend and content recommendations (Aiello et al., 2012), shopping and movie recommendation (Huang et al., 2005), knowledge graph completion (Nickel et al., 2015) and even important social causes such as identifying criminals based on past activities (Hasan et al., 2006). Historically, link prediction methods have utilized topological graph features such as common neighbors yielding strong baselines like Adamic/Adar measure (Adamic & Adar, 2003), Jaccard Index among others. Other approaches include Matrix Factorization (Menon & Elkan, 2011) and more recently deep learning and graph neural networks based approaches (Grover & Leskovec, 2016; Wang et al., 2015; Zhang & Chen, 2018) have risen to prominence. A commonality among all the above approaches is that the link prediction problem is define over a single dense graph where the objective is to predict unknown/future links within the same graph. Unlike these previous approaches, our approach considers link prediction tasks over multiple sparse graphs which are drawn from distribution over graphs akin to real world scenario such as protein-protein interaction graphs, 3D point cloud data and citation graphs in different communities.

In meta-learning or learning to learn (Bengio et al., 1990; 1992; Thrun & Pratt, 2012; Schmidhuber, 1987), the objective is to learn from prior experiences to form inductive biases for fast adaptation to unseen tasks. Meta-learning has been particularly effective in few-shot learning tasks with a few notable approaches broadly classified into metric based approaches (Vinyals et al., 2016; Snell et al., 2017; Koch et al., 2015), augmented memory (Santoro et al., 2016; Kaiser et al., 2017; Mishra et al., 2017) and optimization based approaches (Finn et al., 2017; Lee & Choi, 2018). Recently, there are several works that lie at the intersection of meta-learning for few-shot classification and graph based learning. In Latent Embedding Optimization, Rusu et al. (2018) learn a graph between tasks in embedding space while Liu et al. (2019) introduce a message propagation rule between prototypes of classes. However, both these methods are restricted to the image domain and do not consider meta-learning over a distribution of graphs as done here.

Another related line of work considers the task of few-shot relation prediction in knowledge graphs. Xiong et al. (2018) developed the first method for this task, which leverages a learned matching met-

ric using both a learned embedding and one-hop graph structures. More recently Chen et al. (2019) introduce Meta Relational Learning framework (MetaR) that seeks to transfer relation-specific meta information to new relation types in the knowledge graph. A key distinction between few-shot relation setting and the one which we consider in this work is that we assume a distribution over graphs while in the knowledge graph setting there is only a single graph and the challenge is generalizing to new types of relations within this graph.

## 6  DISCUSSION AND CONCLUSION

We introduce the problem of few-shot link prediction—where the goal is to learn from multiple graph datasets to perform link prediction using small samples of graph data—and we develop the Meta-Graph framework to address this task. Our framework adapts gradient-based meta learning to optimize a shared parameter initialization for local link prediction models, while also learning a parametric encoding, or signature, of each graph, which can be used to modulate this parameter initialization in a graph-specific way. Empirically, we observed substantial gains using Meta-Graph compared to strong baselines on three distinct few-shot link prediction benchmarks. In terms of limitations and directions for future work, one key limitation is that our graph signature function is limited to modulating the local link prediction model through an encoding of the current graph, which does not explicitly capture the pairwise similarity between graphs in the dataset. Extending Meta-Graph by learning a similarity metric or kernel between graphs—which could then be used to condition meta-learning—is a natural direction for future work. Another interesting direction for future work is extending the Meta-Graph approach to multi-relational data, and exploiting similarities between relation types through a suitable Graph Signature function.

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

## 7 APPENDIX

### 7.1 A: EGO-AMINER DATASET CONSTRUCTION

To construct the Ego-Aminer dataset we first create citation graphs from different fields of study. We then select the top 100 graphs in terms number of nodes for further pre-processing. Specifically, we take the 5-core of each graph ensuring that each node has a minimum of 5-edges. We then construct ego networks by randomly sampling a node from the 5-core graph and taking its two hop neighborhood. Finally, we remove graphs with fewer than 100 nodes and greater than 20000 nodes which leads to a total of 72 graphs as reported in Table 1.

### 7.2 B: ADDITIONAL RESULTS

We list out complete results when using larger sets of training edges for PPI, FIRSTMM DB and Ego-Aminer datasets. We show the results for two metrics i.e. Average AUC across all test graphs. As expected, we find that the relative gains of Meta-Graph decrease as more and more training edges are available.

| PPI Convergence | 10% | 20% | 30% | 40% | 50% | 60% | 70% |
|---|---|---|---|---|---|---|---|
| Meta-Graph | **0.795** | **0.831** | **0.846** | **0.853** | 0.848 | 0.853 | 0.855 |
| MAML | 0.745 | 0.820 | 0.840 | 0.852 | **0.854** | 0.856 | **0.863** |
| Random | 0.578 | 0.651 | 0.697 | 0.729 | 0.756 | 0.778 | 0.795 |
| No Finetune | 0.738 | 0.786 | 0.801 | 0.817 | 0.827 | 0.837 | 0.836 |
| Finetune | 0.752 | 0.8010 | 0.821 | 0.832 | 0.818 | **0.856** | 0.841 |
| Adamic | 0.540 | 0.623 | 0.697 | 0.756 | 0.796 | 0.827 | 0.849 |
| MAML-MLP | 0.603 | 0.606 | 0.606 | 0.606 | 0.604 | 0.604 | 0.605 |
| Deepwalk | 0.664 | 0.673 | 0.694 | 0.727 | 0.731 | 0.747 | 0.761 |

Table 5: AUC Convergence results for PPI dataset for training edge splits

| PPI-5 updates | 10% | 20% | 30% | 40% | 50% | 60% | 70% |
|---|---|---|---|---|---|---|---|
| Meta-Graph | **0.795** | **0.829** | **0.847** | **0.853** | 0.848 | **0.854** | **0.856** |
| MAML | 0.756 | 0.837 | 0.840 | 0.852 | **0.855** | **0.855** | **0.856** |
| No Finetune | 0.600 | 0.697 | 0.717 | 0.784 | 0.814 | 0.779 | 0.822 |
| Finetune | 0.582 | 0.727 | 0.774 | 0.702 | 0.804 | 0.718 | 0.766 |
| MAML-MLP | 0.603 | 0.606 | 0.603 | 0.604 | 0.603 | 0.606 | 0.605 |

Table 6: 5-gradient update AUC results for PPI for training edge splits

| FirstMM DB Convergence | 10% | 20% | 30% | 40% | 50% | 60% | 70% |
|---|---|---|---|---|---|---|---|
| Meta-Graph | **0.782** | **0.786** | 0.783 | **0.781** | 0.760 | 0.746 | 0.739 |
| MAML | 0.776 | 0.782 | **0.793** | **0.785** | **0.791** | 0.663 | 0.788 |
| Random | 0.742 | 0.732 | 0.720 | 0.714 | 0.705 | 0.698 | 0.695 |
| No Finetune | 0.740 | 0.710 | 0.734 | 0.722 | 0.712 | 0.710 | 0.698 |
| Finetune | 0.752 | 0.735 | 0.723 | 0.734 | 0.749 | 0.700 | 0.695 |
| Adamic | 0.504 | 0.519 | 0.544 | 0.573 | 0.604 | 0.643 | 0.678 |
| Deepwalk | 0.487 | 0.473 | 0.510 | 0.608 | 0.722 | **0.832** | **0.911** |

Table 7: AUC Convergence results for FIRSTMM DB dataset for training edge splits

| FirstMM DB 5 updates | 10% | 20% | 30% | 40% | 50% | 60% | 70% |
|---|---|---|---|---|---|---|---|
| Meta-Graph | **0.773** | **0.767** | **0.743** | **0.759** | 0.742 | **0.732** | 0.688 |
| MAML | 0.763 | 0.750 | 0.624 | **0.776** | **0.759** | 0.663 | **0.738** |
| No Finetune | 0.708 | 0.680 | 0.709 | 0.701 | 0.685 | 0.683 | 0.653 |
| Finetune | 0.705 | 0.695 | 0.704 | 0.704 | 0.696 | 0.658 | 0.670 |

Table 8: 5-gradient update AUC results for FIRSTMM DB for training edge splits

| Ego-Aminer Convergence | 10% | 20% | 30% | 40% | 50% | 60% | 70% |
|---|---|---|---|---|---|---|---|
| Meta-Graph | **0.626** | **0.738** | **0.786** | **0.791** | **0.792** | **0.817** | **0.786** |
| MAML | 0.561 | 0.662 | 0.667 | 0.682 | 0.720 | 0.741 | 0.768 |
| Random | 0.500 | 0.500 | 0.500 | 0.500 | 0.500 | 0.500 | 0.500 |
| No Finetune | 0.548 | 0.621 | 0.673 | 0.702 | 0.652 | 0.7458 | 0.769 |
| Finetune | 0.623 | 0.691 | 0.723 | 0.764 | 0.767 | 0.792 | 0.781 |
| Adamic | 0.515 | 0.549 | 0.597 | 0.655 | 0.693 | 0.744 | 0.772 |
| Deepwalk | 0.602 | 0.638 | 0.672 | 0.686 | 0.689 | 0.711 | 0.731 |

Table 9: AUC Convergence results for Ego-Aminer dataset for training edge splits

| Ego-Aminer 5 updates | 10% | 20% | 30% | 40% | 50% | 60% | 70% |
|---|---|---|---|---|---|---|---|
| Meta-Graph | **0.620** | 0.5850 | **0.732** | 0.500 | **0.790** | **0.733** | 0.500 |
| MAML | 0.500 | 0.504 | 0.500 | 0.500 | 0.519 | 0.500 | 0.500 |
| No Finetune | 0.500 | 0.500 | 0.500 | 0.500 | 0.500 | 0.500 | 0.500 |
| Finetune | 0.608 | **0.675** | 0.713 | **0.755** | 0.744 | 0.706 | **0.671** |

Table 10: 5-gradient update AUC results for Ego-Aminer for training edge splits

