# OpenReview forum: "Meta-Graph: Few shot Link Prediction via Meta Learning"
_ICLR.cc/2020/Conference — Reject_

### Official Review · AnonReviewer1 · 2019-10-23
**Official Blind Review #1**

**Rating:** 6

**Review:**

This paper presents a new link prediction framework in case of having seen only a small fraction of the graph. The premise is that the predictor is trained on other similar graphs. The predictor (a variational graph autoencoder) performs transfer learning through gradient-based metalearning with a combination of global and local parameters.

The paper is conceptually solid, however, it is hard to claim novelty in individual ideas in the paper. The major concern I have is with the benchmarks. It seems like the (meta) training data is very similar to the validation data. So, even though it is supposed to be a few shot prediction problem, it seems like the meta-training phase has already provided more than a "few" training points. Please correct me if this observation is incorrect. My rating is neutral so far (not an option in the system) and ready to change upon comments from the authors.

**Experience Assessment:**

I have read many papers in this area.

**Review Assessment: Checking Correctness Of Derivations And Theory:**

I assessed the sensibility of the derivations and theory.

**Review Assessment: Checking Correctness Of Experiments:**

I carefully checked the experiments.

**Review Assessment: Thoroughness In Paper Reading:**

I read the paper at least twice and used my best judgement in assessing the paper.

---

> ### Author Response · Authors · 2019-11-14
> **Response to Reviewer 1**
>
> We thank the Reviewer for their valuable feedback and helpful comments regarding issues that could be clarified in the draft. In particular, we now address the reviewers' main concern that the meta-training data is very similar to the validation data. First, we wish to address the fact that our problem definition and setup for few shot link prediction follows standard approaches to few shot classification using Meta-Learning—with Model Agnostic Meta-Learning (MAML) by Finn et. al [1] being a particularly representative example. Taking MAML as an example, we first partition our dataset of tasks/graphs into training and testing splits. Specifically, this means that when evaluating Meta-Graph we test on unseen graphs for fast adaptation similar to how a meta-learner is evaluated on novel classification tasks. Furthermore, for each task/graph G_i we consider meta-training and meta-validation samples, which under a conventional few-shot classification setting is a pair of input data and its corresponding label (i.e., (x_j, y_j)) while in few shot link prediction each sample is an edge, e_j taken from the task/graph. This effectively means that training and validation edges are drawn from the same graph, which is analogous to the classification setting of drawing training and validation samples from the same set of classes for a particular task T_i. Thus by design meta-training and meta-validation data are similar as they are taken from the same graph but as we are in the few-shot setting the amount of meta-training edges available to the local link prediction model is always <30% while the meta-validation edges used to perform higher order gradients is always capped at 20% of edges for a graph. We argue that during meta-training our meta-learner is still under the few-shot setting as it doesn’t have access to a majority of edges for each graph, which is the standard setting for link prediction. Furthermore, when evaluating our approach we initialize a local link prediction model using learned global parameters. Crucially, there is no validation set of edges used in this phase as the global link prediction model does not need to be updated and as a result, the data is partitioned into <30% of edges for training on test graphs while the remaining edges are solely used for evaluation. We believe our setup for few shot link prediction is a natural extension to the few-shot classification via meta-learning setting where each task is a link prediction on a new graph and each sample is an edge from this graph. We highlight in a subsection of section 2: “Relationship to traditional meta-learning” how this new graph setting introduces unique challenges and how our Meta-Graph approach is able to overcome this. We have updated this section with further clarification between these two few-shot settings.
>
> [1]. Finn, Chelsea, Pieter Abbeel, and Sergey Levine. "Model-agnostic meta-learning for fast adaptation of deep networks." Proceedings of the 34th International Conference on Machine Learning-Volume 70. JMLR. org, 2017.

---

### Official Review · AnonReviewer3 · 2019-10-23
**Official Blind Review #3**

**Rating:** 6

**Review:**

Overview: In this paper, a meta-learning approach is proposed to perform link prediction across multi-graphs with scarce data. To do so, each graph is treated as a link prediction "task". Different from the tasks in conventional meta-learning, the graphs here are generally non i.i.d. Based on the variational graph auto-coders, the method learns two important components: the global parameters used for GNNs and a graph signature function used to modulate the parameters of inference model based on the history of observed training graphs.
The idea of formulating link prediction as a few-shot learning problem and solving it via multi-graph meta-learning is novel. The approach basically takes advantage of meta-learning and is expected to generalize well across graphs. The numerical study is extensive in discussing the properties of meta-graph and showing more attractive empirical performance than the existing approaches.


Comments:

1. The explanation of Figure 1 needs to be improved. In its current form, the figure does not illustrate the idea of this paper very clearly. It would be better to provide more explanations on the graphical model for meta-graph and the meta-graph architecture in the context.

2. The description of Algorithm 1 is confusing in some places. For example, the operation in Line 6 is commented as *compute graph signature*. However, what's actually going on is to set the signature function to stop gradient computation which seems not matching the comment.

3. In section 4.2, the 5-gradient update AUC is reported to show the fast adaptation of meta-graph. However, in my opinion, it could be much more informative to show the accuracies of different gradient steps, like what has done in the experimental study of MAML. I don’t think the current experimental results can well explain the property of fast adaptation.

=== update after author response ===

Thank you for your response.   I find my main concerns properly addressed in the feedback.

**Experience Assessment:**

I have read many papers in this area.

**Review Assessment: Checking Correctness Of Derivations And Theory:**

N/A

**Review Assessment: Checking Correctness Of Experiments:**

I assessed the sensibility of the experiments.

**Review Assessment: Thoroughness In Paper Reading:**

I read the paper at least twice and used my best judgement in assessing the paper.

---

> ### Author Response · Authors · 2019-11-14
> **Response to Reviewer 3**
>
> We value the comments made by the Reviewer regarding parts of our submission that were unclear which we now address. We acknowledge that the explanation for the graphical model in its current state is difficult to interpret specifically due to a typo in Eq. 7 which should read $$p(\mathcal{E}|\theta) = \prod_j^J \left(\int \int p(\mathcal{E}_{j}| \phi_j)p(\phi_j|\psi,\tilde{\phi_j})p(\tilde{\phi_j}|\theta)d\phi_j d\tilde{\phi_j}\right)$$. This typo has been corrected in the current draft and we will add more description connecting the graphical model to the architecture diagram. With regards to Line 6 in Algorithm 1, we agree with the reviewer that the comment “compute graph signature” can be misleading when for the local link prediction model we attach a stop gradient computation to prevent the Graph Signature function from being updated in the inner loop of optimization. We have fixed this minor typo in Algorithm 1 in the updated draft and we will add further emphasis in the text to highlight the exact nature of the computation. Finally, with regards to the reviewers' point about 5-gradient AUC being reported in Table 3, we also report in our original draft fast adaptation with different gradient steps for PPI and FIRSTMM DB datasets with 10% of edges. Specifically, Fig 2 contains AUC vs. Gradient Steps of Meta-Graph and other baselines from which we see the utility of the Graph Signature which allows for fast adaptation on different gradient steps. We have further commented on this in the updated version of the paper while highlighting that Table 2 is for three distinct training edge splits —-i.e. 10%,20%,30% for all datasets while Figure 2 is for 10% of training edges on PPI and FIRSTMM DB.

---

### Official Review · AnonReviewer2 · 2019-10-24
**Official Blind Review #2**

**Rating:** 6

**Review:**

This paper proposes to provide a novel gradient-based meta-learning framework (Meta-Graph) for a few shot link prediction task. More specifically, they generate an effective parameter initialization for a local link prediction model for any unseen graph by leveraging higher-order gradients and introducing graph signature function into graph neural network framework. The authors validate the proposed model through several experiments.

This paper reads well and the results appear sound. I personally find the idea of incorporating the structural signature for input graphs into the GCN to modulate the parameters of the inference model very interesting. Moreover, the provided experiments support the authors’ intuition and arguments.

As for the drawbacks, I find the relationship to the prior works partly unclear. Moreover, it would be nice if the authors could also provide some ideas for future research directions, such as the prospects of using their approach for improving link prediction models and incorporating Meta-Graph in other domains like molecules structure. My concerns are as follows:

•    I am wondering if you can adopt R-GCN [1] instead of the GCN model for extending the Meta-Graph to multi-relational graphs?
•    I suggest considering ranking metrics such as MRR and HITS@ to further evaluate the performance of Meta-Graph.
•    Comparing the performance of Meta-Graph and MAML in Table 2, I am wondering about the reason behind the fact that the distance between these models’ performance decreases by increasing the number of edges in PPI and FirstMMDB, but increases in Ego-AMINER? Further, I am wondering why Meta-Graph performance drops in Ego-AMINER after increasing the number of edges from 10% to 20% (Table 3)?
•    I suggest providing a comparison of computational complexity between Meta-Graph and the baselines.

[1] Schlichtkrull, Michael, et al. "Modeling relational data with graph convolutional networks".


**Experience Assessment:**

I have published in this field for several years.

**Review Assessment: Checking Correctness Of Derivations And Theory:**

I assessed the sensibility of the derivations and theory.

**Review Assessment: Checking Correctness Of Experiments:**

I assessed the sensibility of the experiments.

**Review Assessment: Thoroughness In Paper Reading:**

I read the paper thoroughly.

---

> ### Author Response · Authors · 2019-11-14
> **Response to Reviewer 2 Part 1**
>
> We greatly appreciate the reviewers interest in our work in addition to the insightful comments which we now address. The key clarification points are points are outlined below and are grouped by theme.
>
> General Concerns:
> While we distinguish Meta-Graph from traditional Link Prediction and Meta-Learning work in Section 2 of the submission we agree with the reviewer that this discussion of related work can be expanded. In particular, we will add further discussion on related works to link prediction and recent approaches to few-shot learning via meta learning and put our contribution in context. The reviewer also raises a great point regarding the computational complexity of Meta-Graph in relation to other baselines. As Meta-Graph uses higher order gradients, empirically we observe moderate to significant overhead during training time in comparison to Finetuning or NoFinetuning baselines. In contrast to vanilla MAML, the overhead cost is substantially less as the Graph Signature function is updated in the outer loop and it’s gradients are cut off during the training of the local link prediction model. We have added a discussion regarding the computational cost of Meta-Graph and MAML in section 4.1 in the updated version of the submission.
>
> Experimental Results:
> The reviewer makes a great observation that the difference in performance of Meta-Graph and MAML in Table 2 decreases with increasing number of edges in PPI and FIRSTMM DB but increases in Ego-AMINER. We first note that each of these datasets are from vastly different domains from biological networks in PPI, 3D point cloud data in FIRSTMM DB and citation networks. Furthermore, the graph properties —-i.e. (#edges, #nodes, #node features, etc… ) for graphs in each dataset are also differ greatly making fair comparisons for performance metrics between datasets difficult. We speculate that one possible reason for observing an increasing trend in Ego-AMINER is due to the inherent difficulty in optimization using higher order gradients in Meta-Learning [1] which is somewhat alleviated in GS-Gating via a gating mechanism. We have added a note on this particular point in the updated version of the draft.
>
> The reviewer makes another great remark that the performance of Meta-Graph drops relative to other baselines for 20% of edges on the Ego-AMINER dataset after 5-gradient steps as reported in Table 3. However it is important to note that the final convergence result for Meta-Graph under the same setting as reported in Table 2 is higher than all other baselines. We hypothesize that this is due to an artifact of the optimization process which is somewhat unique to Ego-AMINER as even conventional MAML does poorly in the fast adaptation setting. We have added further discussion regarding the unique optimization challenges observed in Ego-AMINER, which warrant further investigation in future work
> .
> With regards to the Reviewers comments on evaluation metrics reported for link prediction we report AUC which is a standard metric reported in both the Variational Graph AutoEncoder (VGAE) by Kipf et. al [2], Node2Vec by Grover et. al [3] and this review paper on link prediction [4]. We agree with the reviewer that ranking metrics such as MRR and HIT@K are important but we argue the results presented using AUC offer a natural source for comparison given that our local link prediction builds of the VGAE architecture.
>
> Part 1/2

---

> > ### Author Response · Authors · 2019-11-14
> > **Response to Reviewer 2 Part 2**
> >
> > Future Research Directions:
> > Our work proposes a general purpose framework for few-shot link prediction using Meta-Learning and a learned graph signature function which can be deployed in any domain such as molecular graphs or recommendation systems. The deployment of Meta-Graph in domain specific settings and further analysing the properties learned by the Graph Signature function in these domains is an interesting direction for future work. In principle, the local link prediction model used in Meta-Graph is not restricted to a regular GCN and R-GCN based model can be used for multi-relational data. An important caveat though is that Meta-Graph is useful for few-shot link prediction and is not built for few-shot relation prediction where the goal is to adapt quickly to novel relations [5] in knowledge graphs. At a high level, Meta-Graph seeks to exploit structural similarities across graphs drawn from a distribution of graph to bootstrap the learning process to an unseen graph from the same distribution and thus multi-relational graphs can also be accommodated given an appropriate choice of a local link prediction model and graph signature function.
> >
> > [1]. Antoniou, Antreas, Harrison Edwards, and Amos Storkey. "How to train your MAML." arXiv preprint arXiv:1810.09502 (2018).
> > [2] Kipf, Thomas N., and Max Welling. "Variational graph auto-encoders." arXiv preprint arXiv:1611.07308 (2016).
> >
> > [3] Grover, Aditya, and Jure Leskovec. "node2vec: Scalable feature learning for networks." Proceedings of the 22nd ACM SIGKDD international conference on Knowledge discovery and data mining. ACM, 2016.
> > [4] Wang, Peng, et al. "Link prediction in social networks: the state-of-the-art." Science China Information Sciences 58.1 (2015): 1-38.
> > [5] Wenhan Xiong, Mo Yu, Shiyu Chang, Xiaoxiao Guo, and William Yang Wang. One-shot relational learning for knowledge graphs. arXiv preprint arXiv:1808.09040, 2018.
> >
> > Part 2/2

---

### Public Comment · ~Jack_Ma2 · 2019-10-31
**Reproduce experimental results**

Dear authors,
          I run the source code via the above link. It is hard to reproduce the experimental results by running the run_ppi_best_gs.sh.  The besyesian optimzation results are already used in this script. I use the gating-based signature function, it is better than the orginal gs-modulation function in eq.(6). However, it is still unable to achieve the experimental results provided in the paper.  Could you please provide more details for reproduction?

---

> ### Author Response · Authors · 2019-10-31
> **Reproducing experimental results**
>
> Hi Jack,
>
> Thanks for your interest in our work and especially for taking the time to go through the code. While it is true that run_ppi_best_gs.sh contains our hyperparameters after Bayesian Optimization this is intended for the GS-Modulation encoder and while the hyperparameters should be similar for other variations of the Graph Signature function there are small differences which affect the final number. The final results can be obtained by running the plotter script but require an wandb username which has been redacted for the review process but will be available during a full public release of the code. We have these hyperparameters and we will update the code repo for GS-Gating at the next allowable opportunity. Further, it is important to note that there is inherent randomness in the codebase even after fixing all possible random seed due to the fact that we use the Pytorch Geometric repo which has some sampling GPU operations that introduce some unavoidable randomness. In spite of this, the final AUC results should be reasonably close to our reported numbers for the same hyperparameter configuration. For more information on GPU randomness in Pytorch Geometric, you can refer to this git issue: https://github.com/rusty1s/pytorch_geometric/issues/92 . Finally, as you have already discovered GS-Gating does better than GS-Modulation which is an important point that we also report in the paper. Thanks again for taking the time to go through the codebase!

---

### Decision · Program_Chairs · 2019-12-19

**Decision:**

Reject

**Comment:**

This paper presents a new link prediction framework in the case of small amount labels using meta learning methods. The reviewers think the problem is important, and the proposed approach is a modification of meta learning to this case. However, the method is not compared to other knowledge graph completion methods such as TransE, RotaE, Neural Tensor Factorization in benchmark dataset such as Fb15k and freebase.  Adding these comparisons can make the paper more convincing.